# Effect of Drought on the Development of *Deschampsia caespitosa* (L.) and Selected Soil Parameters during a Three-Year Lysimetric Experiment

**DOI:** 10.3390/life13030745

**Published:** 2023-03-09

**Authors:** Jakub Elbl, Vojtěch Lukas, Julie Sobotková, Igor Huňady, Antonín Kintl

**Affiliations:** 1Department of Agrosystems and Bioclimatology, Faculty of AgriSciences, Mendel University in Brno, Zemědělská 1, 613 00 Brno, Czech Republic; 2Agriculture Research Ltd., Zahradní 1, 664 41 Troubsko, Czech Republic

**Keywords:** drought, basal respiration, microbial activity, mineral nitrogen, climate change

## Abstract

This work presents results from a field experiment which was focused on the impact of the drought period on microbial activities in rhizosphere and non-rhizosphere soil. To demonstrate the effect of drought, the pot experiment lasted from 2012 to 2015. Fifteen lysimeters (plastic containers) were prepared in our area of interest. These lysimeters were filled with the subsoil and topsoil from this area and divided into two groups. The first group consisted of two variants: V1 (control) and V2 (84 kg N/ha), which were not stressed by drought. The second group consisted of three variants, V3 (control), V4 (84 kg N/ha), and V5 (84 kg N/ha + 1.25 L lignohumate/ha), which were stressed by drought every year of the experiment for 30 days. Changes in the soil moisture content caused by drought significantly affect the growth of *Deschampsia caespitosa* L., the microbial activity, and the soil’s capacity to retain nutrients. The measured basal respiration and dehydrogenase activity values confirm the significant effect of drought on microbial activity. These values were demonstrably higher in the period before drought simulation by more than 60%. On the other hand, significant differences between microbial activities in the rhizosphere and non-rhizosphere soil were not found. We did not find a clear effect of drought on the formation of soil water repellency.

## 1. Introduction

Drought is currently regarded as a new phenomenon, but it is a natural part of the climate [1,2] and an extreme condition of the hydrological process [3,4]. Drought represents a threat to the world food security and its impacts on soil have to be monitored [2,3]. Stressing the soil by drought has consequences for both its production and non-production functions [3,5] which are reflected in the ability to provide nutrients for plants [6] or retain water [7].

Rhizosphere and rhizosphere soil are the most important soil parts where significant processes take place, which affect not only the soil fertility but the entire soil ecosystem, i.e., the ability to retain nutrients, resistance to erosion, or soil filtration capacity [8]. Rhizosphere is a zone of the soil which is affected by the roots of plants. Rhizosphere soil can be defined as a part of the soil profile, which has direct interaction with plant roots [9]. Rhizosphere can be considered as a unique zone of the soil environment, where the exchange of substances occurs between the soil (organisms contained therein) and the plant; based on this interaction, a soil–plant system develops that brings benefits to both parties. The plant provides exudates (carbonaceous substances) while the soil organisms provide secretions (polysaccharides, amino acids → products of cellular metabolism) to the plant. The correct functioning of the system increases the stability of the whole ecosystem. On the other hand, the non-rhizosphere or extra-root soil is missing the above interaction, thus being dependent on the transport of substances via the soil eluate [8,9,10].

The direct impact of drought on microbial activity in the soil environment and consequently also on soil fertility, health, and quality can be expressed by the presence of microbial enzymes in the soil [11] or by the level of soil respiration activity which includes the respiration of organisms and plant roots [12]. Together with the lack of moisture, changes in these parameters then subsequently lead to the stress of plants that have to put limits on the production of biomass, which results in the damage of whole plant communities [13].

The reason for using enzyme activities and soil respiration as indicators of drought impact on the soil environment is the fact that, according to Borken et al. [6] and Sanaullah et al. [11], these parameters are related to selected soil properties such as water and nutrient availability. Microorganisms play a key role in the soil in all terrestrial ecosystems as they mediate a range of ecological processes which represent their building stones. They include, namely, the cycling of nutrients, the decomposition of SOM, or the control and conservation of the biodiversity in vegetation cover [14]. However, the response of organisms to abiotic and biotic factors differs across the soil environment [11,12]. The soil environment has both direct and indirect influence on the soil organisms. Soil properties such as texture or SOM content affect parameters of the soil environment (temperature, moisture content, availability of nutrients, etc.) and hence the environment of microorganisms. The direct influence of conditions in the soil environment can be monitored in soil microorganisms occurring outside of the root system (in a so-called “bulk soil”). Contrariwise, microorganisms in the rhizosphere soil are affected by the soil environment conditions primarily indirectly, through the effect on plants. Plants can be affected, for example, by long periods of drought (when the wilting point is reached in the soil) or by excessive precipitation [15]. Plants react either by restricting or intensifying the production of root exudates which represent the main signaling device between the plant and the microbial symbiotic network in the soil matrix [9,10,15].

The significance of rhizosphere as being a unique environment in which processes take place that are indispensable for the natural functioning of the soil has been confirmed by a number of scientific studies. Rhizosphere soil has a key influence on the soil’s capacity to retain nutrients, namely reactive N (N_r_), i.e., oxidized, reduced, and organic forms of N. If the input of N_r_ into the soil is higher than the soil can utilize (e.g., in the form of mineral N), the soil environment and soil microorganisms become saturated with this N_r_ [16,17]. Having been saturated, the soil microorganisms cannot process (immobilize) any further Nr. If the microbial activities in the soil experience a secondary influence (e.g., fluctuation of soil moisture content due to drought), N-nutrients are lost from the soil, most frequently by being washed out [16]. There is a direct connection between the degree of microbial activity in the soil and the soil’s capacity to retain N_min_ (i.e., N_r_) [16,18] whereas microorganisms are indispensable for making Nr available to plants and for its further utilization in the soil [19].

Changes in microbial activities as well as changes in the composition of microbial communities due to drought may be reflected in the shifted values of soil hydrological limits. The contents of hydrophobic and hydrophilous substances in the soil strongly affect the soil potential of accepting water molecules [20]. Soil water repellence or soil hydrophobicity (SWR) was first characterized in detail in semiarid and subtropical climatic conditions when studying the consequences of fires [21], with the first reference to SWR being published in the American study by Schreiner and Schorey [22]. The principle of hydrophobicity consists of the creation of hydrophobic coats on the soil aggregates thanks to which water is repelled and the wettability of soil is reduced [23]. SWR affects the soil structure, the stability of soil aggregates, and the availability of nutrients to plants via the natural way [24]. According to Mataix-Solera and Doerr [25], soils with an increased content of hydrophobic substances feature increased surface runoff, decreased availability of water for plants, and hence reduced absorption of substances by root systems. The level of SWR can be determined using laboratory and field measurements. Laboratory methods dwell on determining certain spectra of organic substances in the soil sample that are responsible for the formation of the hydrophobicity of soil particles according to the opinions of scientists [24,26]. In addition to the laboratory methods, there are also field methods used to establish the infiltration of various fluids into the soil [27,28]. Another possibility is to use methods of the laboratory measurement of rate and the characteristics of the process of infiltration of the precise amount of water, i.e., a drop of water with a specific volume [26,29].

The aim of this study was to evaluate the direct effect of drought stress on the development of *Deschampsia caespitosa* L. and microbial activity (soil respiration) in rhizosphere and non-rhizosphere soil. Furthermore, the effect of drought was studied on the formation of soil hydrophobicity and the washing out of Nmin from the soil. For the purpose of studying the above-mentioned objective, the following hypotheses were tested: zero hypothesis (H_0_) and alternative hypothesis (H_1_). H_0_: changes in the soil water content caused by extreme climatic phenomena have no influence on the development of the model plant, soil microbial activities, or a loss in nutrients from the soil, do not influence the level of soil hydrophobicity, and their action cannot be mitigated by the method of farmland fertilization. H_1_: changes in the soil water content caused by extreme climatic phenomena adversely affect the model plant development, microbial activities, and a loss in nutrients from the soil. They also show a change in soil hydrophobicity. In the case of agricultural land, the negative effects can be corrected using the method of fertilization.

## 2. Materials and Methods

### 2.1. Field Experiment

The above hypothesis was tested in a lysimetric experiment (Figure 1 and Figure A1) which was established in the protection zone of an underground source of drinking water in Březová nad Svitavou (annual climatic averages 1962–2012: precipitation 588.47 mm and air temperature 7.9 °C; Figure 2). The experiment lasted from September 2012 to October 2015. Fifteen plastic (PVC) pots were used as experimental containers, each of the same size (Figure 1), and filled with 25 kg of subsoil and topsoil (agrochemical parameters, see Table 1). Soil samples (topsoil and subsoil) were collected in our area of interest (Březová nad Svitavou, Figure 1) in August 2012 according to ISO 10381-6:2009 [30]. Lysimeters were buried (Figure A2) in the ground. Details about our area of interest and the experiment design were published by Elbl et al. [18] and Elbl et al. [31].

The period of drought was simulated using plastic rooves. Five experimental variants were prepared to test the effect of drought, which were divided into two groups (Table 2). The first group had no roof and consisted of two variants: control variants without the addition of fertilizers and variant fertilized with mineral nitrogen (N_min_). The other group was at all times covered with a roof once a year (2013, 2014, and 2015) for 30 days during the growing season. This group contained three variants: control, variant fertilized with mineral nitrogen, and variant fertilized with mineral nitrogen combined with lignohumate (LG).

Treatments V2 and V4 were fertilized with the mineral fertilizer DAM 390 (registered in CZE in line with EU conditions, reg. no. E100), a liquid nitrogenous fertilizer with the content of 30% (w) nitrogen in various forms, of which ½ was amide-N, ¼ was ammonium-N, and ¼ was nitrate-N. These variants were applied using 60% of the recommended N_min_ dose for extensive grass stands, i.e., 84 kg N/ha. Treatment V5 was fertilized with the combination of mineral fertilizer and auxiliary preparation Lignohumate B (LG B; registered in CZE in line with EU conditions, reg. no. R8422), a mixture of humic and fulvic acids and their salts. Based on the recommendation of the manufacturer, a dose for garden lawns was chosen, watering after the establishment, i.e., 12.5 mL/100 m^2^. This dose was completed with the above-mentioned dose of N_min_. Dates of fertilizer applications are presented in Table A1.

### 2.2. Measurement of Basal Respiration

Basal respiration (BR) was measured according to Keith and Wong [34] using soda lime granules as cumulative CO_2_ production for 24 h in a 7-day period, three times before and after drought simulation. The soda lime granules were stored in a plastic container. Before use, the container was opened and inserted into a metal ring. Each lysimeter had two metal rings (see Figure 3 and Figure A3; part A, B). One was located within the rhizosphere zone and the other one was located outside it. The air tightness of this connection was secured with water applied into double metal grooves (see Figure A3; part C). After 24 h, the soda lime granules were taken out from behind the metal sheet cover and inserted into a plastic container which was airtight closed. The soda lime granules were then transported to be weighed in the laboratory. The difference between the original weight and the new weight was directly proportional to the amount of bound CO_2_ (Equation (1)). The results of cumulative CO_2_ production were expressed in g of C per m^2^ and day.

The calculation of basic respiration based on the weight increment of natrocalcite according to the Keith and Wong [34] equation is as follows:(1)$$\mathrm{BR}(g\;C/m^{2}\;24\;h)=\left\{\frac{\left[\left(m_{2}-m_{1}\right)-m_{3}\right]\times 1.69}{S}\right\}\times \left(\frac{24}{t}\right)\times \left(\frac{12}{44}\right)$$

BR was measured every year in three cycles; one cycle always consisted of a min. of two dates of natrocalcite application (Table 3). Each application lasted for 24 h, and after which the individual sample containers with natrocalcite were closed and removed from the lysimeters. The measurement was then repeated after 7 days. When the cycle was ended, the sample containers were transported into the laboratory where the soda lime granules were weighed on analytical scales (ABS-N, KERN & SOHN GmbH, GER). Different dates or absence of measurements were due to unfavorable weather conditions (heavy rain, etc.). For the purposes of the presented study, the first cycle of measurements was at all times implemented before the drought simulation and measurements of the third cycle were taken after the drought simulation.

### 2.3. Determination of Dehydrogenase Enzymatic Activity

Dehydrogenase activity (DHA) was determined using triphenyltetrazolium chloride (TTC) as an acceptor of charged particles released via the oxidation of organic compounds, in accordance with the original method developed by Casida et al. [35]. According to this method, 3 g of fresh (field moist) soil was weighed into a glass tube and mixed with 0.04 g of CaCO_3_, 1 mL of TTC, and then 2.8 mL of distilled water. The prepared samples were incubated at 37 °C for 24 h. After this procedure, triphenyl formazan (TPF—product of the reaction) was extracted using 60 mL of methanol and the resultant solution was filtered. The presence of TPF in the filtrate was indicated by the intensity of a reddish color which was determined using a spectrophotometer (Hach Lange DR 2800, Hach Lange GmbH, Wien, Austria) at the wavelength of 485 nm. DHA was expressed as µg of TPF per 1 g of dry soil matter.

### 2.4. Measuring the Loss in Mineral Nitrogen from the Soil

The loss in N_min_ from the soil was monitored through its leaching in the soil eluate (SOEL) and quantified on the basis of its concentration in SOEL. The amount of SOEL leaked from the lysimeters was measured directly in the control shaft (Figure A2) and recorded. At each SOEL leakage, an average sample (100 mL) was collected from the catching container. The N_min_ concentration in the collected eluate was determined using the distillation–titration method according to Peoples et al. [36] and then converted to 1 L of leaked SOEL. The distillation–titration method [36] was developed to determine the concentration of ammonium and nitrate N leachates and extracts. The analysis was broken down into two separate parts (to establish the concentration of ammonium N first and then the concentration of nitrate N). Ammonium ions were established when the bound ammonium N in the form of NH_3_ was released, which was then cooled down and reacted with the mixture of boric acid and methylene blue as the indicator. The intermediate product that rose after the cooling was NH_4_^+^, which immediately reacted with boric acid. The capture quantity of NH_4_^+^ was determined by titrating HCl at a concentration of 0.0179 N. Prior to the distillation, the nitrate ions were reduced by the addition of Devard alloy (a mixture of 50% Cu, 45% Al, and 5% Zn) to the solution. The product of the reaction is NH_3_, which was determined according to the same principle as ammonium ions.

The total amount of N_min_ was calculated as a sum of these basic N_min_ forms. The analysis was performed on the distillation–titration instrument Behr S3 Stream Distillation Unit (Behr Labor Technik, Düsseldorf, Germany).

### 2.5. Determination of the Degree of Soil Hydrophobicity

Changes in the content of hydrophobic and hydrophilous substances responsible for soil water repellence (SWR) are expressed in the presented study using the value of unsaturated hydraulic conductance (*Kr*; cm/s) as an indirect indicator of SWR [23,26]. The *Kr* value was calculated based on measuring cumulative water infiltration (*I*; mL). The field measurements of I were taken using a Mini Disk Infiltrometer (MDI; Decagon Devices, Inc., Pullman, WA, USA; Figure A4) according to Robichaud et al. [28]. Based on the conducted verification experiments according to Elbl et al. [18] and Elbl et al. [37], tension (*h*_0_) was adjusted to the value of 2 cm. MDI was always placed on the soil surface so that maximum contact with the soil could be ensured. Prior to starting the measurements, the initial water volume in the instrument and at the time of *t* = 0 was recorded; then, the MDI was placed on the measured surface. The loss in water amount was recorded in regular time intervals selected with respect to the rate of infiltration from 30 s to 120 s. The minimum number of readings at each measuring point was 12. The measurements were taken before and after drought simulation in each experimental year in the rhizosphere and non-rhizosphere zones of soil. *K_r_* was calculated according to Lichner et al. [27], Zhang [38], and Lichner et al. [39] using a modified version of Equation (2), originally constructed by Zhang [38].
(2)$$K_{r}\left(h_{0}\right)=\frac{c_{2}\left(h_{0}\right)}{A_{2}}$$

### 2.6. Statistical Analysis

The measured data of individual parameters were assessed using exploratory data analysis to verify the homogeneity and normality of their selection. Potential differences in the activity of dehydrogenase and basal respiration were analyzed using the analysis of variance (one-way ANOVA) in combination with the post-hoc Tukey honestly significant difference (HSD) test. Furthermore, regression analysis was performed to analyze the correlations of selected parameters. All significant differences were detected at *p* < 0.05. All statistical analyses were performed using the program Statistica version 12 CZ (StatSoft, Dell Software, Round Rock, TX, USA).

## 3. Results

### 3.1. Plant Biomass Production

The production of plant biomass (expressed in DM) was variable both within the individual years of the experiment and between the years (Table 4). The highest biomass production in each experimental year was recorded in Variant V2 which was not stressed by drought: 2013—76.72 g; 2014—59.93 g; and 2015—47.53 g. This variant was fertilized with 84 kg N/ha. In 2013 and 2015, the above-mentioned values were not statistically significant as compared with the unfertilized control variant (V1) without drought stress. On the other hand, the values of plant biomass production in Variant V2 (2013, 2014, and 2015) were significantly higher than in the drought-stressed variants (V3, V4, and V5).

In 2013 and 2015, the control variant without drought stress (V1) exhibited significantly higher values of plant biomass production than the control variant stressed by drought (V3). In 2013–2014, the application of fertilizers in the drought-stressed variants (V4—84 kg N/ha; V5—84 kg N/ha + 1.25 L LG B/ha) resulted in a demonstrably higher production of plant biomass as compared with the control variant stressed by drought (V3). However, the values were at all times demonstrably lower than those in the fertilized variant with no drought stress V2. In general, all experimental variants reached the demonstrably highest production of plant biomass during the first year of the experiment (2013).

### 3.2. Basal Respiration

Basal respiration was monitored in the rhizosphere and non-rhizosphere soil before and after drought simulation in three years (2013–2015; Table 5 and Table 6; Figure 4). The measured values are divided into two periods: (I) before and (II) after drought simulation. This also relates to Variants V1 and V2 that were not exposed to drought stress, but BR was measured in them on the same dates as in the drought-stressed variants V3, V4, and V5. Table 5 provides an overview of BR values measured in the rhizosphere and non-rhizosphere soil in the selected periods of the 3-year lysimetric experiment from 2013 to 2015 that would be difficult to evaluate comprehensively without a detailed analysis of the respective periods (Table 6). The highest BR values were recorded in 2013 and 2015, at all times in the first period (first cycle of measurements), i.e., before drought simulation, and always in the rhizosphere soil. The lowest BR values were always recorded in the period after drought simulation, even in Variants V1 and V2 which were not affected by roofing. Moreover, after drought simulation, significant differences in the values of BR were recorded between the rhizosphere and non-rhizosphere soil.

The course of BR values must be further described in the respective experimental years. Year 2013 (Table 5) spots the trend for the whole period of monitoring microbial activity in the R and NR soil during the lysimetric experiment in 2013 and 2015. The highest RS values were demonstrably measured in the period before drought simulation and the lowest ones were measured after drought simulation. Comparing the BR values in the N and NR soil, we can see that there were no significant differences observed in the period before drought simulation (Table 6). On the other hand, differences detected in the period after drought simulation were significant only partly. As compared with the control variant stressed by drought V3(R), the control variant with no drought stress V1(R) exhibited in the period of BR measurement after the drought simulation had the highest values of microbial activity in the rhizosphere soil with the difference between the variants being more than 100%. Furthermore, as compared with the non-rhizosphere soil and its control variant stressed by drought V3(NR), the Variant V1(R) exhibited demonstrably higher BR values.

Compared to the BR values recorded in 2014, the BR values measured in 2015 show a similar course as those recorded in 2013. In all variants, significant differences (Table 6) were detected between the periods with the already mentioned decrease in BR values between Variants V1 and V2 being detected again. For comparison, the BR values in Variants V1(R) and V2(R) after drought simulation were approximately 40% lower compared with 2013 and 2014. The course of BR values in the individual variants in 2014 was different than in 2013 and 2015 (Table 6) as no significant differences were found in the variants affected by drought stress between the periods before and after drought stress simulation. In contrast, a demonstrable increase in BR was detected in the non-rhizosphere soil of Variants V3, V4, and V5, as compared with the rhizosphere soil of these variants in the period before drought simulation. In the period after drought simulation, only minimum differences were recorded. Not even in one of the periods of the experiment in 2015 were significant differences found in the level of microbial activity between R and NR soil in the individual variants.

In general, microbial activity can be evaluated using the average cumulative CO_2_ production (BR). Average BR values for three years of the experiment (2013–2015) are presented in Figure 5. Comparing the respective variants, we found out that the levels of BR were similar in both the non-rhizosphere and rhizosphere soil with variants exposed to drought simulation exhibiting demonstrably higher BR values in the period before the drought simulation by more than 60%. For example, the drought-stressed control variant (V3) exhibited in the NR soil a BR value of 3.79 g CO_2_-C/m^2^∙24 h before drought simulation and of 1.45 g CO_2_-C/m^2^∙24 h after drought simulation. Similarly, variants only exposed to the impact of natural weather conditions (V1–V2) exhibited a demonstrably higher level of BR (4.61 and 5.28 g CO_2_-C/m^2^∙24 h) in the first period than in the second period (2.2 g CO_2_-C/m^2^∙24 h). Thus, a BR decrease by more than 50% occurred in the second period of measurement again.

### 3.3. Dehydrogenase Activity

Similarly, as in the case of BR, DHA was always measured in two periods before and after drought simulation (Figure 5), i.e., in the period before and after the roofing of selected variants. The determination of DHA started in 2014. During the measurements in this year, differences were found among the respective variants in the period before drought simulation (I) as well as in the period after drought simulation (II).

If we focus only on the period before drought simulation (I) in 2014, we can see that the DHA values ranged from 9.38 to 20.62 µg TPF/g∙h, without making a difference between the R or NR soil. Significant differences were recorded between the variants that were not stressed by drought (V1 and V2) as compared with the other variants (V3–V5) stressed by drought both in 2014 and already in 2013. Other differences were detected in the values of DHA between the R and NR soil of variants with no drought stress (V1 and V2). Compared with the NR soil, DHA was about 60% higher in the R soil. In contrast, differences in DHA between the R and NR soil of other variants were not detected.

A similar behavior of values was also apparent in the second period of 2014, i.e., after drought simulation (II), where the same significant differences were recorded as in the preceding period. As compared with the other variants (V3, V4, and V5), the highest level of enzymatic activity was demonstrably exhibited by V1(R soil) and V2(NR soil). The drought-stressed variants exhibited no significant differences even when comparing the R and NR soil.

In 2015, the most significant differences were determined again in the period before drought simulation (I) (Table 7). The range of enzymatic activity was similar as in 2014, i.e., from 10.30 to 20.53 µg TPF/g∙h. The highest activity of microorganisms was once again detected in the R variants with no drought stress in V1—unfertilized control (16.79 µg TPF/g∙h), V2—application of 84 kg N/ha (20.53 µg TPF/g∙h), and also in one drought-stress variant, Variant V3 (16.32 µg TPF/g∙h). These values were significantly the highest as compared with the drought-stressed variants V3 (unfertilized control) and V5 (84 kg N/ha + 1.25 L LG B/ha). Furthermore, significant differences were detected in the values of DHA in NR soil as compared with R soil by about 60% in Variant V1, 55% in Variant V2, and 62% in Variant V5.

The second period of DHA monitoring (II), i.e., after drought simulation in 2015, showed that microbial activity decreased in the R soil of all drought-stressed variants. Among other things, a significant decrease in DHA values was recorded in Variant V2 (NR soil) which was not stressed by drought. The demonstrably highest DHA values were once again detected in the R soil of Variants V1 and V2 as compared with the remaining variants. Furthermore, significant differences in DHA between the R and NR soil were recorded in Variants V1, V2, and V5. In the variants stressed by drought, a significant effect of drought on DHA decrease was recorded as being more than 60%.

### 3.4. Leaching of Mineral Nitrogen

The leaching of mineral nitrogen through the soil eluate (SOEL) was monitored for three years (2013–2015). Although the experiment was already established in October 2012, we had to wait until the end of the year to let conditions in the lysimeter stabilize so that actual soil conditions could be simulated as faithfully as possible. The concentrations of N_min_ in SOEL before and after drought simulation are shown in Figure 6.

In 2013, the highest losses in N_min_ (>15 mg/L) were recorded before drought simulation. The lowest values were detected after drought simulation and ranged around 5 mg/L. Significant differences among the variants were observed only in the period after drought simulation (Table 8). The significantly lowest loss in N_min_ was recorded in the control variant (V1 = 1.90 mg/L) and the second lowest one was in the variant with 84 kg N/ha (V2 = 3.67 mg/L). These variants were not exposed to drought stress, and the loss in N_min_ recorded in them was approximately 60% lower than in the drought-stressed variants. Differences among the drought-stressed variants were detected only between Variant V4 (application of 84 kg N/ha) and Variants V3 (control) and V4 (combined application of N_min_ and C_org_).

In 2014 and 2015, the lowest concentration of N_min_ was again detected in SOEL from Variant V1, both in the period before drought simulation and in the period after drought simulation. It is important to follow differences (a) among the variants within the individual groups of the experiment (V1 vs. V2, V3 vs. V4, etc.), and (b) between the groups (stressed and unstressed by drought). Compared with the unfertilized controls, variants fertilized with 84 kg N/ha (V2 and V4) exhibited that N_min_ losses increased by min. 20% both in the group unstressed by drought (V1) and in the drought-stressed group (V3). Variant V5 fertilized with a similar N dose as V2 and V4, but with the addition of LG (1.25 l/ha), exhibited that demonstrably lower values of N_min_ leached from the soil as compared with Variant V4 (in the period after drought stress in 2013, 2014, and 2015). Moreover, this variant reached the same level of N_min_ loss in 2014 as Variant V2 which was not stressed by drought and Variant V3 (unfertilized control) which was stressed by drought.

Furthermore, a relation was analyzed between the level of microbial activity in the soil and the loss in N_min_ from the soil using regression analysis (Table 9) in order to describe potential interactions between the impact of drought on the soil environment. A negative dependence was detected among the levels of microbial activity for the period of drought stress, expressed by the BR value and N_min_ concentration in SOEL in 2013 (R = −0.67; *p* < 0.05) and 2015 (R = −0.68; *p* < 0.05). The results of the regression analysis are significant and indicate that the lowest level of microbial activity was found in variants with the highest concentrations of N_min_ in SOEL in the rhizosphere soil. The intertwining of individual measured parameters is illustrated by the scheme in Figure 7; the method of fertilization (application of N_min_ and C_org_) and weather conditions affect the production of plant biomass (Table 4), with variants unstressed by drought and enjoying enough nutrients exhibiting the highest values. Furthermore, the application of N_min_ and C_org_ combined with weather conditions significantly affects microbial activity in the soil (Figure 4 and Figure 5); it is, however, impossible to distinguish whether the microbial communities in the rhizosphere and non-rhizosphere soil react differently, as only partial differences were recorded between these zones in the values of BR and DHA. Apart from the above, a negative correlation was also found between the development of microbial activity (Table 9) and the leaching of N_min_ from the soil, which was affected among other things by total precipitation amounts in the individual years of the experiment and by the method of fertilization (Figure 6 and Table 8). The decreased microbial activity resulted in the increased loss in N_min_ from the soil and vice versa. In addition, higher total precipitation amounts combined with the drought simulation led to the increased leaching of N_min_ from the soil.

### 3.5. Unsaturated Hydraulic Conductivity—Expression of the Degree of Soil Water Repellence

Table 10 provides an overview of calculated *K*_r_ values including the representation of significant differences. The measurement of cumulative infiltration and the subsequent calculation of *K*_r_ were made, similarly as in the case of DHA, always in two conditions: (I) before drought simulation, i.e., in the first half of the model plant growing season, and (II) after drought simulation, i.e., in the second half of the model plant growing period. Measured values are therefore divided into these two groups even in the variants (V1 and V2) that were exposed to the impact of natural weather conditions only. In the first year of the experiment (2013), no significant differences were found among the individual variants either before or after drought simulation, nor were any significant differences detected between the periods before and after drought simulation within the individual variants.

Statistically significant differences were detected only in the second year of measurements (2014), in the period after drought simulation. The lowest values of *K_r_* indicating an increased degree of SWR were found in the R soil of variants stressed by drought V3(R)–V5(R) in the period after drought simulation. The highest value was recorded in the NR soil of Variant V5(NR), which was significant in relation to V2(NR) and V3(R). No significant differences were found within the individual variants between the rhizosphere and non-rhizosphere soil. In this period, significant differences were recorded only between the rhizosphere and non-rhizosphere soil across the variants, for example between V3(R) and V2(NR). Thus, the degree of SWR was affected primarily by soil moisture content (Table A3) which was higher in Variant V2 (w = 10.19%) than in Variant V3 (w = 2.44%).

A similar course of values was also observed in 2015 when significant differences were found in the period before drought simulation only between Variants V1(NR), V5(NR), and V2(NR). Variants V1(NR) and V5(NR) exhibited the highest values of *K_r_*. In the second period of 2015, i.e., after drought simulation, the lowest *K_r_* values were detected in the rhizosphere soil of drought-stressed variants (V3–V5) as compared with the control Variant V1 and the variant with applied N_min_. In this period, the absolute highest *K_r_* value was recorded in Variant V1(NR). The only significant difference between the two periods of measurements (I and II) was found in Variant V2 in which a significant increase in the *K_r_* value was detected. No significant differences in the *K_r_* values were found in the individual variants between the periods before and after drought simulation.

*K_r_* values were also evaluated for the whole 3-year period of the lysimetric experiment (Figure 8). Similarly, as in the individual experimental years, the values measured in 2013–2015 showed an increased degree of variability and hence a lower presence of significant differences. Although the *K_r_* values reached in the non-rhizosphere soil of individual variants were lower than in the rhizosphere soil, the differences were non-significant. Similarly, the increase in *K_r_* in Period II after drought simulation was non-significant.

## 4. Discussion

A prerequisite in the presented experiment (Figure 9) was that air temperature increasing due to climate change would result in increased transpiration and evaporation from the soil surface and the leaves of plants, i.e., in the drought. The phenomenon would then affect humidity and the formation of precipitous clouds causing short, intensive rains followed by periods of drought again. Changes in the distribution of precipitation amounts would initiate dry periods of varying lengths. These periods of drought would influence the microbial activity in the rhizosphere and non-rhizosphere soil differently. The dry periods would further act indirectly on increased soil hydrophobicity and increased loss in N_min_ from the soil. Therefore, for the purpose of studying the above-mentioned objective, the following hypotheses were tested: the zero hypothesis (H_0_) and the alternative hypothesis (H_1_). These hypotheses are discussed in detail in the below sub-chapters.

### 4.1. Plant Biomass Production

The production of plant biomass was monitored from 2013 to 2015. In each experimental year, it was affected by the application of N_min_, which significantly increased the biomass production of the model plant. The highest biomass yield was always recorded in Variant V2 (no drought stress) and Variant V4 (drought stress), where 84 kg N/ha was applied (Table 4). The positive response of grass stands to the supplementation of N_min_ in the sense of increased plant biomass production is generally known and properly characterized [16,18,40]. The finding was therefore not surprising. Much more interesting was the influence of drought on the production of plant biomass.

*Deschampsia caespitosa* L. is one of the species frequently occurring in natural grass ecosystems in the EU [40,41] and this is why it was chosen as a model plant for the determination of drought impact on its development and microbial community in its root system. Based among other things related to the wide range of habitats of its occurrence (spring areas, pastures, meadows, etc.), an assumption can be formulated that findings about the effect of drought and method of fertilization on the development of the model plant and its biomass production can be applicable to other grass species as well. The model plant can be defined as being readily adaptable to different soil conditions (edaphically tolerant) and representing sites with available moisture [42]. The model plant also responded in the same way during the presented lysimetric experiment. Although it exhibited decreased plant biomass production during drought simulation, it managed to survive the 30-day periods of drought stress. The sensitivity to drought stress leading to a massive drop in biomass production and the vitality of *Deschampsia caespitosa* L. was also corroborated by other research studies [13,43].

It is also very important to mention that the decrease in the production of plant biomass was demonstrably lower in 2013 and 2014 than in the unfertilized control. This indicates that the application of N_min_ alone (V4) or in combination with C_org_ in the form of humates (V5) acted positively on the mitigation of drought impact. The application of fertilizers was the second most important factor which affected the development of *Deschampsia caespitosa* L. and mitigated the negative effect of drought. It follows that the H_0_ hypothesis was disproved and the alternative H_1_ hypothesis was confirmed. Thus, if we wanted to adapt grass ecosystems to climate change, we would have to increase their species diversity and adopt a suitable regime for the nutrition of these stands. This means supplementing N-substances and ensuring the supply of C_org_. Similar conclusions were also arrived at by Zhang et al. [13]. The findings are important, for example, in areas where the grassing of arable land is necessary to protect surface and subsurface sources of drinking water where there is the degradation of existing soil ecosystems, but also where newly developed grass stands occur due to climate change. This state subsequently leads, among other things, to the leaching of nutrients from the soil environment into the sources of drinking water and to the worsening of rain precipitation infiltration into the soil, by which the replenishment of drinking water supply is put into jeopardy [44].

### 4.2. Basal Respiration

The data of BR presented in Table 5 and Figure 4 indicate a direct impact of drought on microbial activity in the R and NR soil in 2013 and 2015. On the other hand, no effect of drought was found in 2014. According to Waldrop and Firestone [45], soil microbial communities are closely related to the life manifestations of above-ground plant communities. The relation is governed by the content of nutrients in the soil, namely by C- and N-substances, and by the availability of water in the soil, which directly affects the activity of microorganisms [46]. At the beginning of the growing season or during its first half, Variants V2, V4, and V5 were fertilized for each year of the experiment (Table A1) that resulted in the increased presence of nutrients in the soil and boosted the growth of plants. Favorable moisture and temperature conditions on the experimental site at the beginning of vegetation development (Figure 2) positively affected the development of the microbial community in the soil, which was interrupted by drought simulation when significant drops occurred both in 2013 and 2015 (Table 5 and Table 6) and in terms of overall average for three years of the experiment (Figure 4).

Therefore, the BR values indicate a repeated effect (simulation) of drought on the microbial activity in the soil, which showed a pronounced decrease in BR values. The effect of drought on the decrease in microbial activity in the soil was confirmed [47]. On the other hand, a drop in BR values was also recorded in variants not exposed to drought stress (V1 and V2). This could be related to the gradual depletion of nutrients available in the soil after the application of fertilizers and hence could slow down the process of mineralization of SOM which has a direct influence on the metabolic activity of soil microorganisms [47,48]. This drop was, however, not sharp as in the case of drought-stressed variants.

It should also be taken into account that total precipitation amounts during the second half of the vegetation period (Figure 2) were much lower in June and July than in April and May. Habekost et al. [49] point to the fact that dynamics of microbial activity in the soil are variable during the vegetation season depending on the availability and quality of organic resources in the soil, which can be mineralized. This finding would explain values measured in variants unstressed by drought. It is, however, interesting that the authors describe in their research the effect of arable land grassing on the microbial community in the soil. During their observations, they measured the highest BR values during and at the end of vegetation, not at the beginning. The above facts are not in line with the results of BR measurements in V1 and V2; the measured data suggest that the decrease in microbial activity in the given variants occurred during the later vegetation period. The difference was statistically significant in all years of the experiment both in the R and NR soil. According to Bloem et al. [12] and Engelhardt [50], the intensity of BR reflects the amount and quality of C in the soil or in SOM. Thus, the BR value can represent the potential of soil biota (microorganisms) to degrade both native and introduced organic substances because the activity of soil microorganisms increases with the increasing BR value and hence their engagement in biochemical processes increases. This would support our assumption that BR was affected by the availability of nutrients (C, N, etc.) in the soil. Unfortunately, we cannot corroborate this fact on the basis of chemical soil analyses.

As mentioned above, there is a presumption that the records of microbial activity were due to the local meteorological situation. Total precipitation amounts for the first cycle of BR measurement in 2013, 2014, and 2015 were 111.3 mm, 88.8 mm, and 31.3 mm, respectively. Total precipitation amounts measured in the third cycle of BR measurement (after drought simulation) in 2013, 2014, and 2015 were 99.5 mm, 95.5 mm, and 18.2 mm, respectively. BR values after the third cycle of measurements in variants without roofing (V1 and V2) were at a similar level in 2013 and 2014, but a clear drop was recorded in 2015. According to Bloem et al. [12] and Engelhardt [50], such fluctuations in total precipitation amounts could have been the main reason affecting the microbial activity in the soil.

Not even in one experimental period in the individual years (Table 6) were significant differences found in the degree of microbial activity between the R and NR soil of specific experimental variants. An important factor for the clarification of the absence of significant differences between the R and NR soil is again the total precipitation amounts as their fluctuations significantly affect the soil ecosystem and hence the microbial community [12,50]. If we compare total precipitation amounts in the period of 31 days before drought simulation, we shall find a difference between 2015 (32 mm) and 2013 (110.8 mm). The difference points to the situation with enough soil moisture available to soil microorganisms, particularly in 2013, which supported their development when combined with increased temperature (compared with the long-term standard + 1.14 °C; Figure 2) [51,52] as well as their drought resistance [53,54] being roofed both in the R and NR soil. The effect of soil moisture availability was of primary importance for the BR values as compared with the effect of fertilization, and in the case of its shortage, the microbial activity in the soil was adversely affected, particularly in 2013 and 2015. Nevertheless, significant differences in the effect of drought on microbial activity in the R and NR soil were not found.

### 4.3. Dehydrogenase Activity

Dehydrogenases belong to basic microbial enzymes and are referred to as enzymes of the respiratory chain [54,55]. The main significance of these enzymes is in the biological oxidation of organic substances [35] and this is why DHA was selected as a suitable indicator of the effect of drought on microbial activity. The values of DHA were determined ex situ in the collected soil samples, always before and after drought simulation via roofing. DHA was measured in the last two years of the experiment (2014 and 2015). In 2014, significant differences were found among individual variants in the periods before (I) and after (II) drought. There were, however, no significant differences in DHA values between the periods before and after drought simulation. Similarly, Liang et al. [56] did not find any effect of supplemented N_min_ on microbial activity in the soil expressed by DHA compared with the control variant; it is, however, necessary to point out that the origin and character of nutrients play a decisive role. In contrast, Luo et al. [57], for example, inform us that there is a demonstrable effect of N application on DHA in the soil; however, if an application of farmyard manure is compared with the application of mineral fertilizer, the increase will always be to the benefit of the variant with the organic form. On the other hand, Luo et al. [57] and Chu et al. [58] point to the fact that the application of N in the form of N_min_ always increased values of all tested enzymatic activities in the soil. Contrariwise, Jahangir et al. [59] found out that increased application doses of N lead in conventional farming to the reduced content of microbial C in the soil and hence to decreased DHA.

Based on the values measured before and after drought simulation in 2015, it can be deduced that DHA could have been affected by the method of fertilization. The reason for this is a situation whereby the highest values of DHA in the individual periods were determined in Variant V2 fertilized with mineral fertilizer. When the other values recorded in the following period after drought simulation are included, the effect of the stress of the soil and plants due to a shortage of monture is apparent. Results from 2014 and 2015 indicate not only the actual effect of drought on the enzymatic activity in the soil, but also its long-term negative effect on the overall microbial activity in the soil, because in both experimental years the lowest DHA values were measured, which reflect the overall potential oxidation activity of microorganisms in the soil [60] in variants that were exposed to the effect of drought each year from the establishment of the experiment. Based on their 15 years of monitoring the effect of fertilization on enzymatic reactions, Liang et al. [56] inform us that the enzyme of dehydrogenase is present only in the live cells of soil microorganisms, which can be in the latent state. This supports an assumption that DHA can be used as a biological indicator of long-term external effects (climate, management methods, etc.). Other scientific studies such as by Luo et al. [57], Chu et al. [58], and Nielsen [61] point to the potential influence of mineral and organic–mineral N forms on the increase in DHA in spite of the fact that it may be negligible, e.g., as compared with organic N forms. Hueso et al. [53] observed the negative influence of drought on the degree of microbial activity in the soil expressed by DHA, too.

### 4.4. Leaching of Mineral Nitrogen

Based on the presented values of the concentration of N_min_ in SOEL, a combined influence of drought, meteorological phenomena (primarily precipitation, Table 2), and the method of fertilization on the loss in N_min_ from the soil can be assumed. This fact was due to increased precipitation amounts recorded in February–April 2013 (102.5 mm) compared with the same period in the following years (2014 = 73.5 mm; 2015 = 70.1 mm). The increased precipitation caused the formation and leakage of soil eluate from individual lysimeters and hence the increased leaching of N_min_ from the soil in the individual lysimeters. The amounts of captured SOEL are presented in Table A2. In the first experimental year (2013), the values of N_min_ concentration in SOEL were probably affected by the process of the stabilization of biochemical reactions in the soil of the buried lysimeter. Nendel et al. [62] inform us that lysimeter stabilization has a direct influence on the course of N mineralization, the availability of N in the lysimeter soil, and on the capacity of this soil to retain N substances. Lysimeters were filled in September 2012 and the first collections of SOEL were made in February 2013. Thus, the stabilization of lysimeters took approximately four months, which, however, did not have to be sufficient and the amount of leached N_min_ could have been affected by its content in the collected homogenized soil and insufficient formation of optimal soil structure. This fact is also confirmed by the high values of ±SE (Figure 6) indicating great variance in the values of core file both in comparison with the second period of measurement in 2013 (after drought simulation) and with the following years of 2014 and 2015.

As mentioned above, the measured values of N_min_ concentration showed more stable behavior in 2014 and 2015. This fact confirms once again the assumption that the increased leakage of N_min_ in the first period of 2013 was due to the meteorological situation (increased total precipitation amount, Figure 2) and was due to the process of lysimeter stabilization. Variants that were not exposed to drought exhibited lower losses in N_min_ than drought-stressed variants. Different values of N_min_ loss between the individual groups (drought stress and no drought stress) of the experiment confirm the assumed hypothesis of the mediated effect of drought on the loss in N_min_, which was also corroborated by Bimüller et al. [7]. An indicator of the drought effect is the demonstrated increase in N_min_ concentration in the captured SOEL in individual variants (V3–V5) in the period of drought stress because the increased N_min_ concentration was always demonstrated in the preceding period in 2013–2015.

Various scientific works such as those by Bloem [12], Sutton [16], and Kintl et al. [32] claim the loss in nutrients from the soil to be an indicator of disturbed microbial activity in the soil and a disturbed organo-mineral complex of the soil. Particularly, Sutton [16] considers losses in N_r_ from the soil to be not only evidence of damage to soils but also evidence of negative anthropogenic action on the soil ecosystem. This state is also indicated by BR values that were the lowest in variants with the greatest loss in N_min_ in the periods of drought stress in 2013 and 2015, and exhibited negative dependence on the values of N_min_ loss from the soil. The mentioned authors Bloem [12], Sutton [16], and Kintl et al. [32] confirm the effect of the action of adverse external factors on the loss in N_min_ from the soil and its reduced availability to plants.

The presented results further indicate the significance of the fertilization effect on retaining N_min_ in the soil, as well as a possibility to mitigate the impacts of drought by the application of supporting substances, namely those which can support microbial activity in the soil and the resistance of plants to drought stress. The method of fertilization and type of applied fertilizer were other factors which affected the loss in N_min_ from the soil as variants fertilized by DAM 390 fertilizer—84 kg N/ha (V2 and V4)—showed that N_min_ losses increased by min. 20% as compared with unfertilized controls both in the group unstressed by drought (V1 vs. V2) and in the drought-stressed group (V3 and V5 vs. V4).

Furthermore, it is necessary to mention that Variant V5 that was fertilized with the same N dose as V2 and V4 but with the addition of LG (1.25 l/ha) exhibited demonstrably lower values of N_min_ leaching from the soil. Thus, the values measured in Variant V5 indicate a positive influence of the combined application of DAM 390 and Lignohumate LG B on the reduction in N_min_ leasing from the soil. The LG B fertilizer that was applied to Variant V5 represents a liquid humic preparation with the content of salts of humic and fulvic acids (concentration 12%) in a ratio of 1:1. Simplified, this preparation can be an artificially prepared, partly dissolved organic matter. Based on its composition, LG B is likely to affect processes in the rhizosphere, particularly fulvic acids, which it contains and which readily dissociate and become involved in the exchange of C and O_2_, thus representing, according to Zsolnay [63], a potential source of energy for soil microorganisms. This could also explain the negative dependence between BR and N_min_ loss from the soil detected using regression analysis. Variants with a low loss in N_min_ demonstrably exhibited the highest level of microbial activity in 2013 and 2015, and a negative dependence on the N_min_ loss (Table 9). The variants were those unstressed by drought and the variant with the application of LG B. According to Podznyakov et al. [64], preparations based on lignohumate mitigate adverse abiotic factors and positively affect biological activity and the nitrogen balance of the soil. This is also confirmed by values measured in our experiment (Figure 4 and Figure 6).

### 4.5. Soil Water Repellence

Cumulative infiltration (*I*; mL) was measured and unsaturated hydraulic conductance (*K_r_*; cm/s) was calculated in the conditions of lysimetric experiment in 2013–2015, with the measurements made only in the soil zone affected by the root system of the model plant (R soil) in 2013. In 2014 and 2015, the measurements were also taken in the soil zone not directly affected by the root system of the model plant (NR soil). The reason for this was a modified methodology and the extension of monitored parameters, i.e., potential difference in the formation of SWR between the rhizosphere (R) and non-rhizosphere (NR) soil (Figure 8).

The *K_r_* value was used to express the level of SWR as based on scientific studies and experiments of Cosentino et al. [29], Wang et al. [65], Robichaud [66], and Diamantopoulos et al. [67]. It can be stated that parameters expressing the state of soil hydraulic properties can be directly affected by SWR intensity. A negative correlation between the level of SWR and the soil’s capacity to infiltrate water was confirmed by Robichaud et al. [28] and Lichner et al. [27]. More precisely, according to Robichaud et al. [66], the volume and rate of water infiltration into the soil decrease with the increasing SWR, which is subsequently shown by decreased *K_r_* values [27,39].

In the first year of measurements (2013), the measured *K_r_* values were characterized by the absence of significant differences both in the period before drought simulation and in the period after drought simulation. In the following year (2014), significant differences were recorded only in the period after drought simulation. The *K_r_* value was with great probability affected by soil moisture content (Table A3), which greatly oscillated among the individual variants. Its influence on SWR and soil environment capacity to infiltrate and conduct water was confirmed, for example, by Diamantopoulos et al. [67] and Mao et al. [68]. According to Mao et al. [68], the presence of higher amounts of hydrophobic organic compounds and the lower content of soil moisture lead to greater water repellence; this suggests that the persistence of SWR is primarily determined by the interaction between organic compounds and water molecules on the nanoscale. The higher value of soil moisture in 2014 thus indicated that soil pores were filled with water, which impaired the infiltration capacity of the soil (measured lower *K_r_* values). On the other hand, the course of *K_r_* values in 2014 (period after drought stress) signaled a decreased rate of water infiltration into the rhizosphere. Thus, the state indirectly indicates increased hydrophobicity in the soil affected by roots. The presented results suggest that there is a relation between physical–chemical soil properties and the level of SWR, which was confirmed by Mataix-Solera et al. [25]. The significance of hydrological processes in the soil, namely moisturizing and drying out, in the acceleration of SWR formation was further confirmed and characterized by Bodí et al. [69].

In the last experimental year (2015), only partial significant differences in *K_r_* values were found among the individual variants again. Based on Mao et al. [68], Schrama and Bardgett [70], and Sándor et al. [71], there was an assumption that drought simulation (decreased soil moisture content and increased soil temperature) would reset in the lesion of microbial cells and the release of organic substances from the roots (response of plants to drought stress). The two processes should have caused an increased SWR level. Drought-stressed variants really showed a regular decrease in *K_r_*_,_ indicating increased SWR in the upper layer of the soil, but it was not significant with the preceding period before drought simulation. The reason for this was probably a short time of drought stress to which model plants were exposed. The significance of the length of drought periods on changes in the SWR formation and in the distribution of SWR-causing substances was confirmed by Mao et al. [68] and Sándor et al. [71]. It is therefore not possible to reject hypothesis H_0_ on the basis of measured *K_r_* values because the effect of drought on the formation of SWR was not statistically demonstrated in all experimental variants.

## 5. Conclusions

The presented paper is to contribute to studying the effect of drought on the growth of the model plant *Deschampsia caespitosa* L., microbial activities in the rhizosphere and non-rhizosphere soil, and N_min_ loss from the soil (soil capacity to retain this key nutrient). The main goal of the study was a general evaluation of the direct effect of drought on the plant–soil complex and the indirect effects of drought on the degree of soil hydrophobicity. In the performed lysimetric experiment, a zero hypothesis (H_0_) and an alternative hypothesis (H_1_) were formulated: H_1_: changes in the soil water content caused by extreme climatic phenomena adversely affect the model plant development, microbial activities, and loss in nutrients from the soil. They also show changed soil hydrophobicity. In the case of agricultural land, the negative effects can be corrected by the method of fertilization. The hypothesis was confirmed in terms of drought effect on the model plant, microbial activity in the soil, and loss in N_min_ from the soil, and based on which the effect of drought on the selected soil parameters was confirmed, however with different statistical probability and measured intensity of the effect. In the case of the effect of drought on the formation of soil hydrophobicity, no influence was demonstrated and H_1_ was rejected.

Based on the research results, it is possible to state that changes in the soil moisture content caused by drought significantly affect the growth of *Deschampsia caespitosa* L., microbial activity in the soil, and the soil’s capacity to retain nutrients. The measured BR and DHA values confirm the significant effect of drought on microbial activity in the soil both from a current and long-term point of view. The results of the field lysimetric experiment confirm the high significance of drought effect on microbial activity in both rhizosphere and non-rhizosphere soil. Our lysimetric experiment demonstrated only partly that adverse impacts of drought can be eliminated using the application of fertilizer with C available in the form of lignohumates.

Furthermore, a demonstrable influence of drought was detected on the loss in Nmin from the soil environment. Drought-stressed variants always exhibited the loss in Nmin via leaching as being greater than variants unstressed by drought. Based on the above measured data, it is possible to state that the loss in Nmin in the field experiment was affected by the combined action of drought and the fertilization method with the application of DAM mineral fertilizer, with the content of N_min_ itself having a negative influence on the loss in mineral N forms from the soil. In contrast, the application of LG B fertilizer in the period after soil stress by drought demonstrably reduced the loss in N_min_ as compared with the control variant that was stressed by drought as well.

## Figures and Tables

**Figure 1 life-13-00745-f001:**
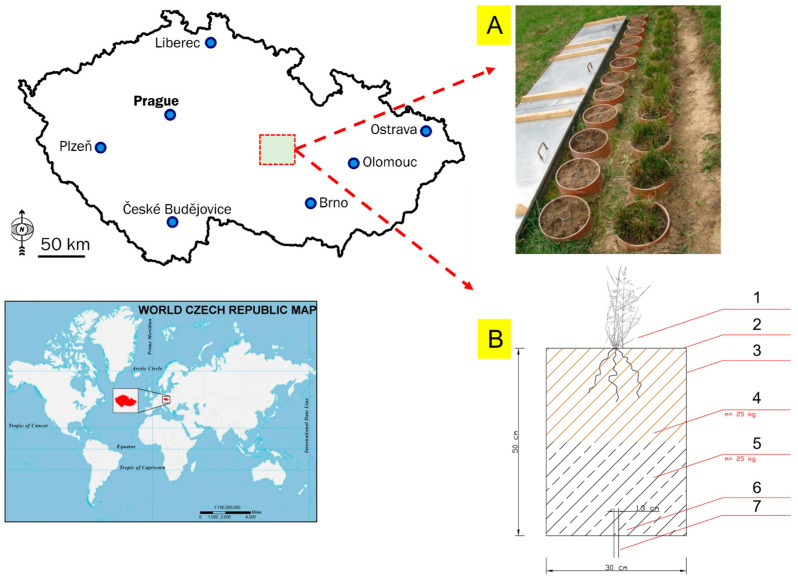
Lysimeter and experimental container. (**A**)—overall view of experimental site; (**B**)—scheme of experimental lysimeter (modified according to Kintl et al. [32]; source of map data: mapsopensource.com). (1) indicator plant—*Deschampsia caespitosa* L. (2) Lysimeter surface area, S = 0.07 m^2^. (3) Experimental container. (4) Sod layer, m = 25 kg. (5) Subsoil layer, m = 25 kg. (6) Drain hole, d = 2.5 cm. (7) Plastic hose.

**Figure 2 life-13-00745-f002:**
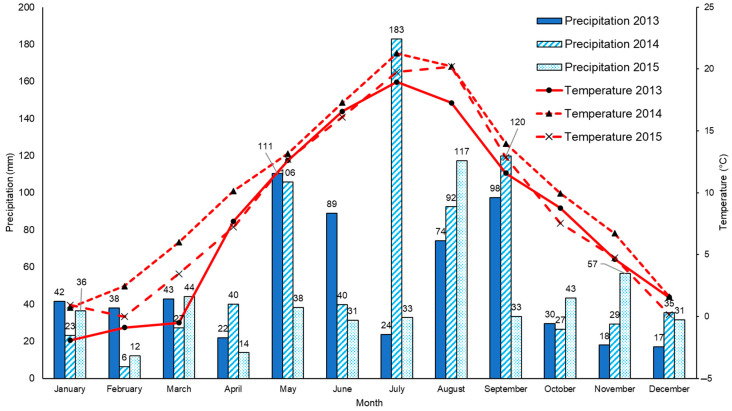
Monthly precipitation amounts and average monthly temperatures measured in the area of interest during the lysimetric experiment (2013–2015).

**Figure 3 life-13-00745-f003:**
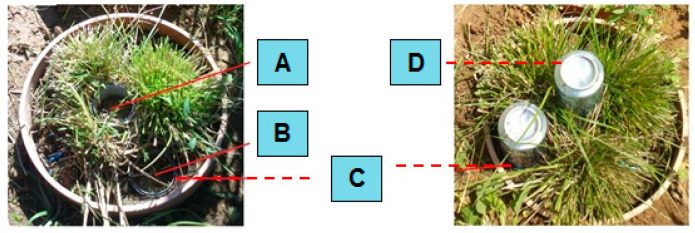
Monthly precipitation amounts and average monthly temperatures measured in the area of interest during the lysimetric experiment (2013–2015). (**A**) soil with the influence of the root system. (**B**) soil without the influence of the root system. (**C**) metal ring for attaching the aluminum cover. (**D**) aluminum cover—natrocalcite is placed under the cover.

**Figure 4 life-13-00745-f004:**
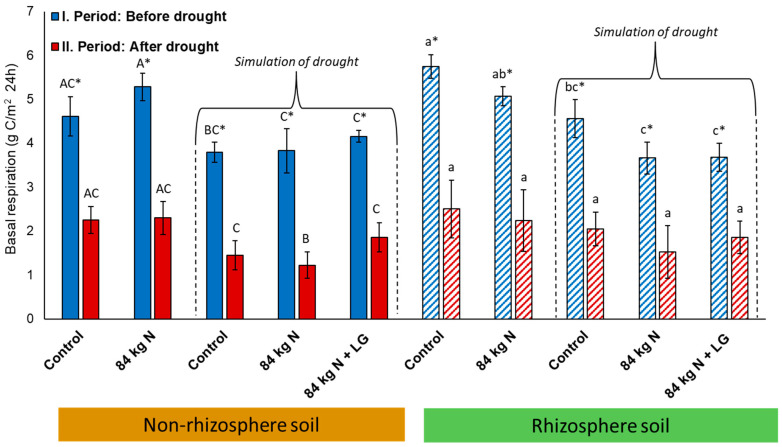
Average cumulative CO_2_ (g CO_2_-C/m^2^ ∙ 24 h) production from rhizosphere and non-rhizosphere zones of soil in lysimetric experiment, 2013–2015. Different capital letters confirm HSD in the non-rhizosphere soil and different lowercase letters confirm HSD in the rhizosphere soil. The differences always relate to a specific period (I or II). Symbol * illustrates a significant difference within a particular variant between the periods before and after drought simulation. Different bars show the drought-stressed variants.

**Figure 5 life-13-00745-f005:**
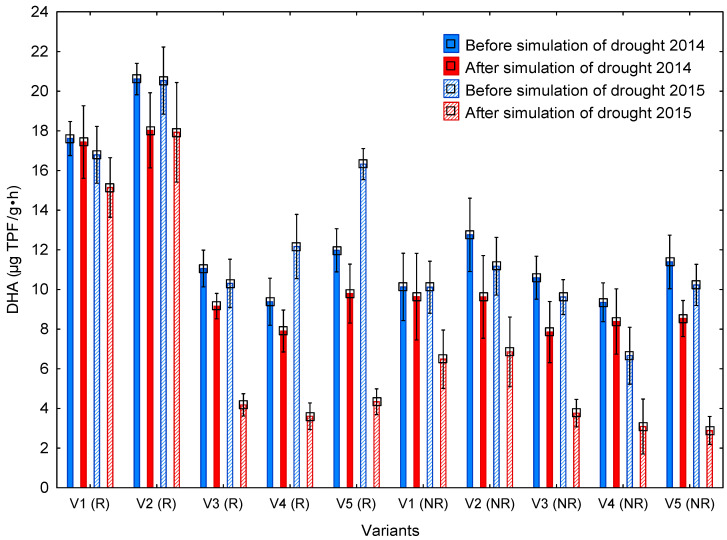
Dehydrogenase activity in soil samples before and after drought simulation. The graph presents the average values of DHA (µg TPF/g∙h; n = 3) which were subjected to ANOVA at a significance level of *p* < 0.05; error bars represent ± SD. Legend: V1 = control; V2 = 84 kg N/ha; V3 = control—stressed by drought; V4 = 84 kg N/ha—stressed by drought; V5 = 84 kg N/ha + 1.25 l LG B/ha—stressed by drought; R = rhizosphere soil; and NR = non-rhizosphere soil.

**Figure 6 life-13-00745-f006:**
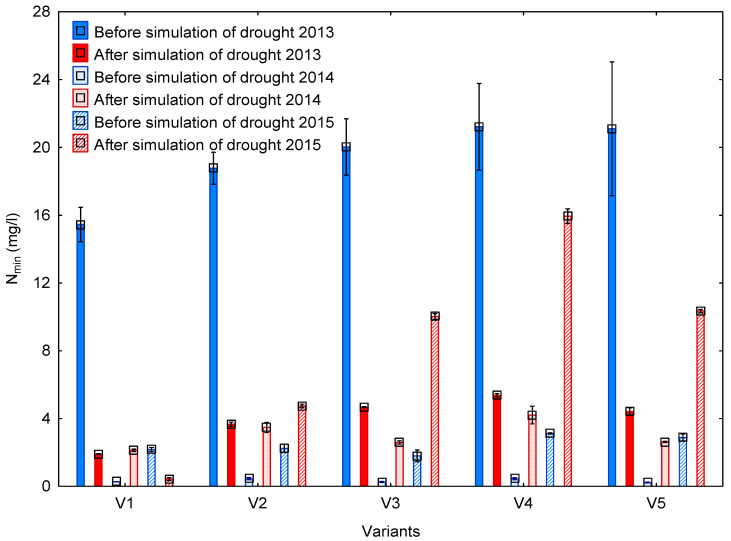
Concentrations of N_min_ captured in the soil eluate. The graph presents the average values of N_min_ loss (mg/L; n = 3). The individual values were subjected to ANOVA at a significance level of *p* < 0.05; error bars represent ± SD. Legend: V1 = control; V2 = 84 kg N/ha; V3 = control—stressed by drought; V4 = 84 kg N/ha—stressed by drought; V5 = 84 kg N/ha + 1.25 L LG B/ha—stressed by drought; R = rhizosphere soil; and NR = non-rhizosphere soil.

**Figure 7 life-13-00745-f007:**
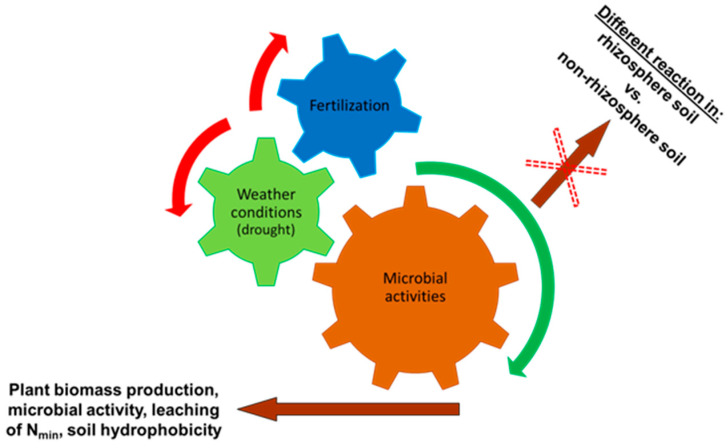
Interaction between individual experimental parameters and external influences.

**Figure 8 life-13-00745-f008:**
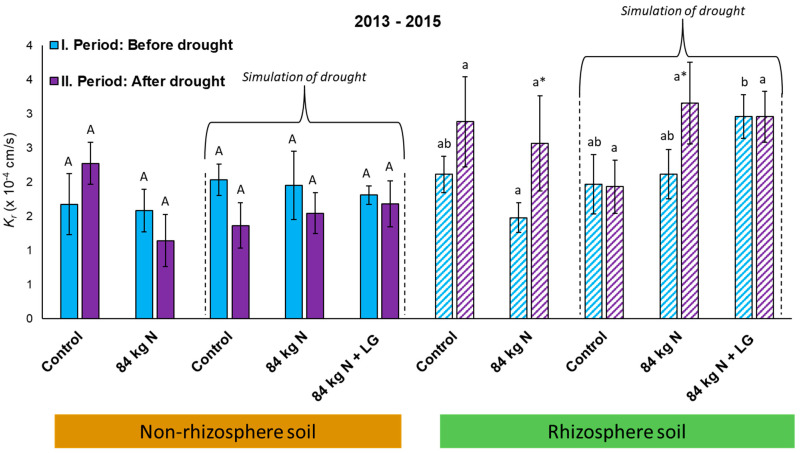
Average *K_r_* values (cm/s) in the rhizosphere and non-rhizosphere soil of the model plant, 2013–2015. The graph presents average *K_r_* values (cm/s; n = 9) ± SD. Different capital letters confirm HSD in the non-rhizosphere soil and lowercase letters in the rhizosphere soil. The differences are always related to a specific period (I. or II.). The * symbol represents a significant difference within a specific variant between the periods before and after drought simulation. Different bars show the drought-stressed variants.

**Figure 9 life-13-00745-f009:**
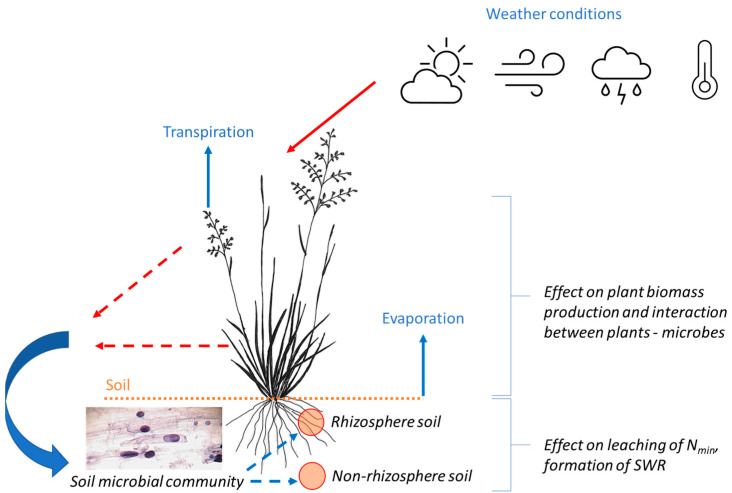
Graphical illustration of the working hypothesis and theoretical basis of the implemented lysimetric experiment.

**Table 1 life-13-00745-t001:** Basic agrochemical characteristics—the plant’s available nutrient content of topsoil and subsoil used for establishment of experiment (soil type: fluvisol; textural class: sandy loam soil).

Soil Sample	mg/kg				K:Mg	pH (H_2_O)	pH (H_2_O)
	P	K	Ca	Mg			
Topsoil (0–0.25 m)	148	343	3596	192	1.78	5.75	6.3
Subsoil (0.25–0.40 m)	225	98	2843	71	1.38	6.15	6.7

Legend: Contents of individual nutrients were established using Mehlich III extractant [33]. Atomic absorption spectrometry was used to establish the content of available potassium (K), magnesium (Mg), and calcium (Ca), while the content of available phosphorus in extract was established calorimetrically.

**Table 2 life-13-00745-t002:** General overview of the lysimetric experiment.

Variants	Group	Characteristic
V1	No roof—without simulation of drought	Control
V2	No roof—without simulation of drought	84 kg N/ha
V3	Roof—simulation of drought	Control
V4	Roof—simulation of drought	84 kg N/ha
V5	Roof—simulation of drought	84 kg N/ha + 1.25 LG B/ha

**Table 3 life-13-00745-t003:** BR measurement dates and numbers of cycles.

Measurement	2013		2014		2015	
	Application	Weighing	Application	Weighing	Application	Weighing
First cycle	21 May		23 April		29 April	
	28 May		5 May		6 May	
	5 June		12 May		12 May	
	12 June	18 June	19 May	21 May	26 May	29 May
Second cycle	3 July		26 May		9 June	
	10 July		5 June		16 June	
	16 July		-		24 June	
	31 July	3 Aug	-	11 June	29 June	8 July
Third cycle	28 August		23 June		13 July	
	4 September		1 July		22 July	
	10 September		8 July		-	
	25 September	3 Oct	14 July	5 Aug	-	7 Aug

**Table 4 life-13-00745-t004:** Above-ground plant biomass production.

Variants	2013			2014			2015		
	Dry Matter (g)	± SD	HSD	Dry Matter (g)	± SD	HSD	Dry Matter (g)	±SD	HSD
V1	73.31 *	1.52	AC	33.53	1.71	AC	45.12	2.69	A
V2	76.72 *	1.63	A	59.93	2.26	B	47.53	1.06	A
V3	57.70 *	1.57	B	29.56	2.08	C	29.47	1.04	B
V4	68.93 *	1.06	C	41.80	2.37	A	33.14	0.95	B
V5	66.81 *	2.20	C	41.34	1.18	A	29.62	1.22	B

Legend: V1 = control; V2 = 84 kg N/ha; V3 = control—stressed by drought; V4 = 84 kg N/ha—stressed by drought; V5 = 84 kg N/ha + 1.25 L LG B/ha—stressed by drought. The table presents the average values of the production of shoot biomass DM of the model plant (g; n = 3) ± SD. Green tinted variants were not stressed by drought but rather exposed to natural weather conditions only. Different capital letters indicate HSD between individual variants within individual years. Symbol * confirms HSD in plant biomass production between the first year of the experiment (2013) and the following years (2014 and 2015). All HSDs were established at a significance level of *p* < 0.05; ANOVA; post hoc HSD Tukey test.

**Table 5 life-13-00745-t005:** Above-ground plant biomass production.

Variants	2013				2014				2015			
	Before Simulation of Drought		After Simulation of Drought		Before Simulation of Drought		After Simulation of Drought		Before Simulation of Drought		After Simulation of Drought	
	BR	±SD	BR	±SD	BR	±SD	BR	±SD	BR	±SD	BR	±SD
V1(R)	5.40	0.66 a	3.02	0.92 a	4.01	0.42 abc	2.07	0.12 ab	4.42	0.82 a	1.67	0.47 a
V2(R)	6.20	0.38 a	2.65	0.29 abc	4.40	0.30 abc	2.95	0.41 ab	5.25	0.78 a	1.29	0.49 a
V3(R)	4.53	0.31 a	1.32	0.03 b	2.59	0.17 b	2.01	0.25 ab	4.26	0.17 a	1.01	0.09 a
V4(R)	5.14	0.19 a	1.63	0.13 abcd	1.66	0.11 b	1.73	0.28 b	4.69	0.25 a	0.30	0.16 a
V5(R)	6.32	0.71 a	2.24	0.11 abcd	1.83	0.12 b	2.32	0.19 ab	4.31	0.07 a	1.02	0.13 a
V1(NR)	4.59	0.54 a*	1.41	0.22 abcd	6.60	1.98 a*	3.83	0.16 a	6.06	0.54 a*	2.26	0.25 a
V2(NR)	4.58	0.41 a	1.40	0.19 abcd	4.71	0.48 abc	3.68	0.80 a	5.92	0.8 8a	1.62	0.29 a
V3(NR)	4.84	0.54 a	0.91	0.15 c	3.22	0.17 c	3.56	0.35 a	5.63	0.37 a	1.68	0.42 a
V4(NR)	4.27	0.05 a	1.10	0.09 d	1.77	0.04 c	2.46	0.40 a	4.95	0.48 a	1.03	0.47 a
V5(NR)	4.13	0.03 a	1.50	0.10 abcd	1.84	0.20 c	3.07	0.33 a	5.07	0.60 a	1.00	0.39 a

Legend: V1 = control; V2 = 84 kg N/ha; V3 = control—stressed by drought; V4 = 84 kg N/ha—stressed by drought; V5 = 84 kg N/ha + 1.25 l LG B/ha—stressed by drought; R = rhizosphere soil; and NR = non-rhizosphere soil. The table presents average values of cumulative CO_2_ production (g CO_2_-C/m^2^∙24 h; n = 3) ± SD from rhizosphere (R) and non-rhizosphere (NR) zones of soil in lysimetric experiment. Different lowercase letters confirm HSD in one year and period (before or after drought simulation). Symbol * indicates HSD in individual variants between the periods before and after drought simulation during one year. All HSDs were established at a significance level of *p* < 0.05 (ANOVA in combination with post hoc HSD Tukey test).

**Table 6 life-13-00745-t006:** Statistical analysis of potential differences in microbial activity (BR) among experimental variants, 2013–2015.

Variants	2013			2014			2015		
	Across Groups Among All Variants (R + NR)		Between Periods “Before and After” in One Variant (R|NR)	Across Groups Among All Variants (R + NR)		Between Periods “Before and After” in One Variant (R|NR)	Across Groups Among All Variants (R + NR)		Between Periods “Before and After” in One Variant (R|NR)
	Before	After		Before	Before		Before	After	
V1(R)	A	A	*	ABC	AB	*	A	AB	*
V2(R)	A	ABD	*	ABC	AB	*	A	AB	*
V3(R)	A	BC	*	B	AB	-	A	AB	*
V4(R)	A	ABCD	*	B	B	-	A	B	*
V5(R)	A	ABCD	*	B	AB	-	A	AB	*
V1(NR)	A	ABCD	*	A	A	-	A	A	*
V2(NR)	A	ABCD	*	ABC	A	-	A	AB	*
V3(NR)	A	C	*	C	AB	-	A	AB	*
V4(NR)	A	D	*	C	AB	-	A	AB	*
V5(NR)	A	ABCD	*	C	AB	-	A	AB	*

Legend: V1 = control; V2 = 84 kg N/ha; V3 = control—stressed by drought; V4 = 84 kg N/ha—stressed by drought; V5 = 84 kg N/ha + 1.25 l LG B/ha—stressed by drought; R = rhizosphere soil; and NR = non-rhizosphere soil. The table presents the results of statistical analysis of data shown in Table 5. Different capital letters confirm HSD between variants in individual periods (before and after drought simulation) and among years (2013, 2014, and 2015). HSD between the periods before and after drought in individual variants and years is marked with the * symbol. All HSDs were established at a significance level of *p* < 0.05; ANOVA with post hoc HSD Tukey test.

**Table 7 life-13-00745-t007:** Statistical analysis of potential differences in microbial activity (DHA) among individual experimental variants, 2014–2015.

Variants	2014			2015		
	Before	After	Difference	Before	After	Difference
V1(R)	A	A	-	A	A	
V2(R)	A	A	-	A	A	
V3(R)	B	B	-	B	B	*
V4(R)	B	B	-	B	B	*
V5(R)	B	B	-	A	B	*
V1(NR)	B	B	-	BC	BC	
V2(NR)	B	B	-	B	B	*
V3(NR)	B	B	-	BC	BC	*
V4(NR)	B	B	-	C	BC	
V5(NR)	B	B	-	BC	C	*

Legend: V1 = V1 = control; V2 = 84 kg N/ha; V3 = control—stressed by drought; V4 = 84 kg N/ha—stressed by drought; V5 = 84 kg N/ha + 1.25 l LG B/ha—stressed by drought; R = rhizosphere soil; and NR = non-rhizosphere soil. The table presents the results of the statistical analysis of data presented in the graph in Figure 5. Different capital letters confirm HSD among the variants in the individual periods (before and after drought simulation) and years (2014 and 2015). HSD between the periods before and after drought in the individual variants is marked as *. All HSD values were determined at a significance level of *p* < 0.05; ANOVA with post hoc HSD Tukey test.

**Table 8 life-13-00745-t008:** Statistical analysis of potential differences in the leaching of N_min_ among the individual experimental variants, 2014–2015.

Variants	2013			2014			2015		
	Before	After	Diff.	Before	After	Diff.	Before	After	Diff.
V1	A	A	*	A	AC	*	AB	A	*
V2	A	B	*	A	BCD	*	AB	B	*
V3	A	C	*	A	C	*	A	C	*
V4	A	D	*	A	D	*	B	D	*
V5	A	C	*	A	C	*	AB	C	*

Legend: V1 = control; V2 = 84 kg N/ha; V3 = control—stressed by drought; V4 = 84 kg N/ha—stressed by drought; V5 = 84 kg N/ha + 1.25 l LG B/ha—stressed by drought; R = rhizosphere soil; and NR = non-rhizosphere soil. The table presents the results of the statistical analysis of data shown in the graph in Figure 5. Different capital letters confirm HSD among the variants in the individual periods (before and after drought simulation) and years (2013, 2014, and 2015). HSD between the periods before and after drought within the individual variants is represented by the * symbol. All HSD values were determined at a significance level of *p* < 0.05 (ANOVA in combination with post hoc HSD Tukey test).

**Table 9 life-13-00745-t009:** Results of regression analysis: BR vs N_min_—regression with N_min_ as dependent variable.

Variants	Range of Values Compared	R	SD from R	SD of Estimate	*p*-Value
Leaching of N_min_ 2013	15	−0.67	0.2051	0.9523	0.0002
Leaching of N_min_ 2014	15	−0.06	0.2768	0.8963	0.8245
Leaching of N_min_ 2015	15	−0.69	0.2034	0.4181	0.0001

**Table 10 life-13-00745-t010:** *K_r_* values (cm/s) in the rhizosphere (R) and non-rhizosphere (NR) soil of the model plant from 2013 to 2015.

Variants	2013				2014				2015			
	Before Simulation of Drought		After Simulation of Drought		Before Simulation of Drought		After Simulation of Drought		Before Simulation of Drought		After Simulation of Drought	
	*K_r_* ± SD (×10^−4^)	HSD	*K_r_* ± SD (×10^−4^)	HSD	*K_r_* ± SD (×10^−4^)	HSD	*K_r_* ± SD (×10^−4^)	HSD	*K_r_* ± SD (×10^−4^)	HSD	*K_r_* ± SD (×10^−4^)	HSD
V1(R)	1.44 ± 0.45	A	1.94 ± 0.38	A	1.57 ± 0.26	A	2.29 ± 0.24	ABC	2.01 ± 0.62	AB	2.59 ± 0.31	AB
V2(R)	2.46 ± 0.41	A	0.72 ± 0.33	A	2.05 ± 0.47	A	2.07 ± 0.56	ABC	0.24 ± 0.05 *	A	2.67 ± 0.25	AB
V3(R)	2.34 ± 0.29	A	2.10 ± 0.49	A	2.33 ± 0.23	A	1.11 ± 0.49	A	1.44 ± 0.17	AB	0.88 ± 0.11	A
V4(R)	2.18 ± 0.62	A	1.81 ± 0.18	A	2.17 ± 0.20	A	1.40 ± 0.32	AB	1.51 ± 0.69	AB	1.48 ± 0.40	AB
V5(R)	1.59 ± 0.27	A	1.17 ± 0.35	A	1.82 ± 0.05	A	1.91 ± 0.19	ABC	2.03 ± 0.10	AB	1.97 ± 0.49	AB
V1(NR)	-	-	-	-	1.58 ± 0.23	A	2.08 ± 0.46	ABC	2.65 ± 0.31	B	3.68 ± 0.87	B
V2(NR)	-	-	-	-	2.37 ± 0.32	A	3.43 ± 0.86	B	0.59 ± 0.12	A	1.70 ± 0.54	AB
V3(NR)	-	-	-	-	2.10 ± 0.59	A	2.46 ± 0.66	ABC	1.84 ± 0.28	AB	1.41 ± 0.13	AB
V4(NR)	-	-	-	-	2.12 ± 0.21	A	3.31 ± 0.13	ABC	2.12 ± 0.51	AB	3.01 ± 1.06	AB
V5(NR)	-	-	-	-	2.96 ± 0.37	A	4.14 ± 0.09	C	2.95 ± 0.26	B	1.77 ± 0.66	AB

Legend: V1 = control; V2 = 84 kg N/ha; V3 = control—stressed by drought; V4 = 84 kg N/ha—stressed by drought; V5 = 84 kg N/ha + 1.25 L LG B/ha—stressed by drought; R = rhizosphere soil; and NR = non-rhizosphere soil. The table presents average Kr values (cm/s; n = 3) ± SD measured in the rhizosphere (R) and non-rhizosphere (NR) soil. Different capital letters confirm HSD in one year and period (before or after drought simulation). Bold *K_r_* values together with the * symbol indicate HSD in individual variants between the periods before and after drought simulation during one year. All HSD values were determined at a significance level of *p* < 0.05 (ANOVA in combination with post hoc HSD Tukey test).

## Data Availability

Experimental data supporting the findings of this study are available from the corresponding authors (J.E. and A.K.) upon request.

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
