# Peer review of "Effect of Drought on the Development of Deschampsia caespitosa (L.) and Selected Soil Parameters during a Three-Year Lysimetric Experiment"

_life, 2023, doi:10.3390/life13030745_

Round 1

Reviewer 1 Report

The paper is perfectly written and a very nice experiment is accomplished. My only concern about the paper is overloaded with lots of work. I would highly recommend to make it a bit brief. This almost 29 pages article.

Moderate English changes required throughout the article.

Author Response

Dear Reviewer,

We thank you for your comments and detailed examination of our paper. All your comments have been included and corrected in the text of the paper.  The explanation is proposed in the text below. We thank for the comments from the reviewer that have contributed to improve the quality of our paper and to eliminate mistakes.

Authors

Reviewer #1:

The paper is perfectly written, and a very nice experiment is accomplished. My only concern about the paper is overloaded with lots of work. I would highly recommend to make it a bit brief. This almost 29 pages article.

Moderate English changes required throughout the article.

Response – The team of authors would like to thank you very much for your feedback. The changes proposed by have been made. We have tried to shorten the introduction, results and discussion of the manuscript.

The authors hope that the revised version of the above manuscript no. life-2224900 will be accepted for publication in the Life journal.

Yours Faithfully,

Authors

Reviewer 2 Report

Drought and the appearance of dry years are a growing problem in the field and grassland management of the northern temperate zone. For this reason, examining and modelling the effect of drought is an important and current task.

The authors have carried out a wide-ranging, multi-faceted series of studies, which is of great importance.

The research design is appropriate, the methods are adequately described.

The presentation of the results is correct, but could be more consistent. Better structured and more visual and informative figures would be needed in order to make it easier to interpret the results. I definitely recommend doing this.

After these minor revisions the manuscript can be published.

Author Response

response to Reviewer' s comments

Dear Reviewer,

We thank you for your comments and detailed examination of our paper. All your comments have been included and corrected in the text of the paper.  The explanation is proposed in the text below. We thank for the comments from the reviewer that have contributed to improve the quality of our paper and to eliminate mistakes.

Authors

Reviewer #2:

Drought and the appearance of dry years are a growing problem in the field and grassland management of the northern temperate zone. For this reason, examining and modelling the effect of drought is an important and current task.

The authors have carried out a wide-ranging, multi-faceted series of studies, which is of great importance.

The research design is appropriate, the methods are adequately described.

The presentation of the results is correct but could be more consistent. Better structured and more visual and informative figures would be needed in order to make it easier to interpret the results. I definitely recommend doing this.

After these minor revisions the manuscript can be published.

Response – The team of authors would like to thank you very much for your feedback. The changes proposed by have been made. We added two schemes to the manuscript to explain:

  1. Interaction between individual parameters.
  2. The theoretical basis of the whole experiment because of which hypotheses were formulated.

The authors hope that the revised version of the above manuscript no. life-2224900 will be accepted for publication in the Life journal.

Yours Faithfully,

Authors

Reviewer 3 Report

I had the opportunity to review the Submission life-2224900. The aim of this study was to evaluate a direct effect of drought stress on the development of Deschampsia caespitosa L. and microbial activity (soil respiration) in the rhizosphere and non-rhizosphere soil. Further, the effect of drought was studied on the formation of soil hydrophobicity and washing out of Nmin from the soil. For the purpose of studying the above-mentioned objective, the following hypotheses were tested: zero hypothesis (H0) and alternative hypothesis (H1). H0: changes in the soil water content caused by extreme climatic phenomena have no influence on the development of model plant, soil microbial activities and loss of nutrients from the soil, do not influence the level of soil hydrophobicity, and their action cannot be mitigated by the method of farmland fertilization. H1: changes in the soil water content caused by extreme climatic phenomena adversely affect the model plant development, microbial activities, and loss of nutrients from the soil. They also show in changed soil hydrophobicity. In the case of agricultural land, the negative effects can be corrected by the method of fertilization.

The paper is an exciting contribution to the field and nicely combines drought with microbial activities to better understand the effect of drought. The paper is very concise and focused.

I do not have any comments and agree to receive the article directly.

Author Response

response to Reviewer' s comments

Dear Reviewer,

We thank you for your comments and detailed examination of our paper. All your comments have been included and corrected in the text of the paper.  The explanation is proposed in the text below. We thank for the comments from the reviewer that have contributed to improve the quality of our paper and to eliminate mistakes.

Authors

Reviewer #3:

I had the opportunity to review the Submission life-2224900. The aim of this study was to evaluate a direct effect of drought stress on the development of Deschampsia caespitosa L. and microbial activity (soil respiration) in the rhizosphere and non-rhizosphere soil. Further, the effect of drought was studied on the formation of soil hydrophobicity and washing out of Nmin from the soil. For the purpose of studying the above-mentioned objective, the following hypotheses were tested: zero hypothesis (H0) and alternative hypothesis (H1). H0: changes in the soil water content caused by extreme climatic phenomena have no influence on the development of model plant, soil microbial activities and loss of nutrients from the soil, do not influence the level of soil hydrophobicity, and their action cannot be mitigated by the method of farmland fertilization. H1: changes in the soil water content caused by extreme climatic phenomena adversely affect the model plant development, microbial activities, and loss of nutrients from the soil. They also show in changed soil hydrophobicity. In the case of agricultural land, the negative effects can be corrected by the method of fertilization.

The paper is an exciting contribution to the field and nicely combines drought with microbial activities to better understand the effect of drought. The paper is very concise and focused.

I do not have any comments and agree to receive the article directly.

Response – The team of authors would like to thank you very much for your feedback.

The authors hope that the revised version of the above manuscript no. life-2224900 will be accepted for publication in the Life journal.

Yours Faithfully,

Authors